# Funders' data-sharing policies in therapeutic research: A survey of commercial and non-commercial funders

**Jeanne Fabiola Gaba**[1,2]*, **Maximilian Siebert**[1,2], **Alain Dupuy**[2], **David Moher**[3,4], **Florian Naudet**[1]

**1** CIC 1414 (Centre d'Investigation Clinique de Rennes), Inserm, CHU Rennes, Univ Rennes, Rennes, France, **2** REPERES (Recherche en Pharmaco-Épidémiologie et Recours aux Soins), CHU Rennes, Univ Rennes, EA, Rennes, France, **3** Center for Journalology, Clinical Epidemiology Program, Ottawa Hospital Research Institute, Ottawa, Canada, **4** School of Epidemiology and Public Health, University of Ottawa, Ottawa, Canada

* fabiolagaba@gmail.com

**Data Availability Statement:** The protocol and its amendments, the extracted data on the funders are available at https://osf.io/ujbf2/.

## Abstract

### Background

Funders are key players in supporting randomized controlled trial (RCT) data-sharing. This research aimed to describe commercial and non-commercial funders' data-sharing policies and to assess the compliance of funded RCTs with the existing data-sharing policies.

### Methods and findings

Funders of clinical research having funded at least one RCT in the years 2016 to 2018 were surveyed. All 78 eligible non-commercial funders retrieved from the Sherpa/Juliet Initiative website and a random sample of 100 commercial funders selected from pharmaceutical association member lists (LEEM, IFPMA, EFPIA) and the top 100 pharmaceutical companies in terms of drug sales were included. Thirty (out of 78; 38%) non-commercial funders had a data-sharing policy with eighteen (out of 30, 60%) making data-sharing mandatory and twelve (40%) encouraging data-sharing. Forty-one (out of 100; 41%) of commercial funders had a data-sharing policy. Among funders with a data-sharing policy, a survey of two random samples of 100 RCTs registered on Clinicaltrial.gov, data-sharing statements were present for seventy-seven (77%, 95% IC [67%-84%]) and eighty-one (81% [72% - 88%]) of RCTs funded by non-commercial and commercial funders respectively. Intention to share data was expressed in 12% [7%-20%] and 59% [49%– 69%] of RCTs funded by non-commercial and commercial funders respectively.

### Conclusions

This survey identified suboptimal performances of funders in setting up data-sharing policies. For those with a data-sharing policy, the implementation of the policy in study registration was limited for commercial funders and of concern for non-commercial funders. The limitations of the present study include its cross-sectional nature, since data-sharing policies

**Funding:** FN received funding from French National Research Agency for this project. Grant Number: ANR-17-CE36-0010 https://anr.fr/ The funder had no role in study design, data collection and analysis, decision to publish, or preparation of the manuscript.

**Competing interests:** I have read the journal's policy and the authors of this manuscript have the following competing interests: FN received funding from the French National Research Agency for this project. This does not alter our adherence to PLOS ONE policies on sharing data and materials.

**Abbreviations:** CSDR, Clinical Study Data Request; DORA, Declaration on Research Assessment; EFPIA, European Federation of Pharmaceutical Industries and Associations; FAIR, Findable, Accessible, Interoperable and Reusable; ICMJE, International Committee of Medical Journal Editors; IFPMA, International Federation of Pharmaceutical Manufacturers and Associations; IPD, Individual Patient Data; LEEM, Les Entreprises du médicament; NIH, National Institutes of Health; OSF, Open Science Framework; PhRMA, Pharmaceutical Research and Manufacturers of America; RCT, Randomized Clinical Trial; SAP, Statistical Analysis Plan; SOAR, Supporting Open Access for Researchers; UK, United Kingdom; USA, United States of America; YODA, Yale University Open Data Access.

are continuously changing. We call for a standardization of policies with a strong evaluation component to make sure that, when in place, these policies are effective.

## Introduction

Ensuring that the science they fund meets the highest research integrity standards is a key issue for funders involved in clinical research. According to the International Committee of Medical Journal Editors (ICMJE), an influential working group of general medical journal editors, trial data-sharing is an ethical imperative [1] and should therefore be one of their priorities. Data-sharing aims to maximize the benefits that can arise from individual patient data (IPD) by exploring new or unresolved issues from completed trials, by pooling them in large IPD meta-analyses or by re-analyzing the initial trial data [1]. It also ensures "transparency, openness, and reproducibility" [2] and evaluates the risk taken by patients participating in clinical trials.

The G7 clearly calls for open practices in science as one of their top priorities [3] and various calls to action from clinical research stakeholders to data generators, such as pharmaceutical companies, universities, charities, regulatory agencies, have led to the implementation of policies and recommendations to responsibly share clinical trial data [4]. Data-sharing platforms such as Clinical Study Data Request, the Yale University Open Data Access (YODA) Project and Vivli were created to facilitate data-sharing and to ensure that clinical trial data are FAIR (Findable, Accessible, Interoperable and Reusable) [5]. The ICMJE has promoted data-sharing by requiring authors to include a data-sharing statement in published articles and to register a data-sharing plan for any new trial [6]. However, to be effective, these initiatives need to be supported by funders, as they require dedicated resources for data-sharing plans of this nature. Data-sharing is not a panacea; it entails considerable challenges, such as privacy and patient consent, and it implies substantial costs for the preparation of the data [7].

Funders are therefore key players in supporting data-sharing and are expected to provide appropriate guidance and to require best practice from their grant recipients [4]. However, funder policies and attitudes toward data-sharing have rarely been explored. Concerning commercial funders, a prior survey of 23 top pharmaceutical companies in 2016 found that almost all of them (22; 96%) had a policy to share IPD. The proportion was 71% among a less selected sample. The policies were however different [8] and lower proportions were found in a 2019 survey with only 25% of large companies making IPD accessible, a proportion that was slightly improved after communication and feedback to the firms [9]. Concerning non-commercial funders, a 2017 survey on 20 non-commercial funders of health research found that 10 had a data-sharing policy with only 2 requiring IPD sharing [10].

We designed this survey to update the previous ones on two large, representative samples of international commercial and noncommercial funders, and to explore the implementation of funders' data-sharing policies. The aims of this research were to evaluate the percentage of funders with a data-sharing policy, to describe these policies and to assess the compliance of funded RCTs with the existing data-sharing policies in terms of intention to share data.

## Methods

A protocol was registered before the start of the research on the Open Science Framework (OSF) (https://osf.io/mkxzf/). This study was divided into two parts: a survey of funder data-sharing policies and a survey of registered RCTs.

All outcomes were reported and described by numbers, percentages and, where appropriate, the corresponding 95% confidence intervals were presented. Verbatim quotes from funder policies were presented qualitatively using examples, word clouds and a detailed list.

For all random samples, we estimated that a random sample of 100 (funders and/or studies) was sufficient to estimate a percentage of 50% (the worst scenario for precision estimates) with a precision (boundaries of the 95 percent confidence interval) of +/- 9.8%.

All analyses were performed with R version 3.4.1.

## Survey of funders' data-sharing policies

**Eligibility criteria and search strategy to identify funders.**    We included funders of clinical research with at least one RCT funded (regardless the design, the population, the intervention or the outcomes) in the course of the years 2016 to 2018, and with an accessible website in English.

We searched for non-commercial and commercial funders. Non-commercial funders were retrieved on the Sherpa/Juliet Initiative website [11]. SHERPA/Juliet is a searchable database providing information on non-commercial funders' policies, especially concerning open access. Commercial funders were selected from different lists of pharmaceutical industry associations: the European Federation of Pharmaceutical Industries and Associations (EFPIA) [12], the Pharmaceutical Research and Manufacturers of America (PhRMA) [13], the International Federation of Pharmaceutical Manufacturers and Associations (IFPMA) [14] and "Les Entreprises du Médicament" (LEEM, a professional organization of pharmaceutical companies operating in France) [15]. We added the list of the top 100 pharmaceutical companies in terms of drug sales in 2016 [16] which was not planned in the first draft of our protocol.

**Funder selection and data extraction.**    A data extraction sheet was developed from a test sample of ten funders that exhibited feasibility of outcome extraction. Two authors (JG, MS) independently performed the eligibility assessment and extracted information from funders' websites. Disagreements were resolved by consensus and a third author (FN) was consulted in case of disagreement.

**Outcomes describing funders' data-sharing policies.**    The primary outcome for this survey was the existence of a data-sharing policy (i.e. clear and explicit documentation). The secondary outcomes concerned the features of the policy: starting date, sanctions in case of non-compliance (and their nature), incentives, type of data shared and documents (IPD, and/or Code, and/or other documents such as protocol, clinical study report or statistical analysis plan), recommended data-sharing platforms (VIVLI, CSDR, YODA, SOAR, etc.), type of data request review panel (independent or not, or mixing independent members and funder members), data request methods (through data-sharing platform, by contacting trial investigators), specific funding for data-sharing, restriction of duration of data availability time frame for sharing data. We classified policies as being "encouraging policies" (policies that mention data-sharing without a strict requirement) or "mandatory policies" (policies that require the implementation of the recommendations and/or that mention sanctions in case of non-compliance). Lastly, the following features for each funder were extracted: country, World Bank income category [17], and whether it is a signatory of the Declaration on Research Assessment (DORA).

## Survey of RCTs funded by funders with a data-sharing policy

**Eligibility criteria and search strategy for RCTs funded by a funder with a data-sharing policy.**    For funders with a data-sharing policy, we examined the practical implementation of the policies. We initially planned to study published RCTs, however, as for most funders the date of implementation of the policy was not reported, it was impossible to judge whether or not any published RCTs was concerned by the data-sharing policy. The protocol was then

amended to focus on data-sharing plans applying to registered RCTs. Any RCT registered after 1st January 2019 was then eligible without any distinction in terms of patients, intervention, comparator or outcome. Two-arm, multi-arm, factorial, cluster, and cross-over trials were included regardless their design (equivalence, non-inferiority or superiority).

RCTs were identified from the Clinicaltrial.gov website. We used the following filters to search for studies to include: the study starting date (after 1st January 2019), study type (interventional studies), funder type (industry filter for commercial sample, NIH and "all other" (a filter that identifies universities, charities and other funders) and filter for the non-commercial sample). Among the RCTs found, we excluded studies where funders were not in our list of funders with data-sharing policies (a study was eligible if it was funded by at least one funder with a data-sharing policy identified in the first survey), and non-randomised studies. Then we selected two random samples of RCTs: 1/ one for non-commercial funders and 2/ one for commercial funders.

**RCT data extraction.**    A data extraction sheet was developed. For each study included two authors (JG, MS) independently extracted the information entered in the "Individual Participant Data (IPD) Sharing Statement" field on the clinicaltrials.gov website. Disagreements were resolved by consensus and a third author (FN) was consulted in case of persistent disagreement.

**Outcomes describing RCT data-sharing statements.**    The primary outcome for this second survey was the intention to share individual participant data in the data-sharing statements of eligible RCTs. The secondary outcomes were the features of the data-sharing statements: data-sharing plan in the registration, information about supporting material availability, information about the protocol and/or statistical analysis plan and/or clinical study report availability, data access methods, restrictions to data access, existence of a specific aim for data reuse, time frame for data availability, free accessibility of data.

## Changes to the initial protocol

As stated above, we modified the methodology for assessing the compliance of RCTs with the sharing policies of their funders and focused on registered data-sharing plans in clinicaltrials. gov. This was done in accordance with the recommendations of ICMJE which states that "clinical trials that begin enrolling participants on or after 1 January 2019 must include a data-sharing plan in the trial registration".

We also made additional minor changes. When information was not publicly available on funder websites, we did not contact them to confirm absence or presence of a policy. Indeed, we previously carried out a similar study on French funders [18] in which the information on data-sharing policies was collected via a questionnaire with email reminders and/or calls in case of non-response. In this survey, we encountered difficulties in collecting the missing information because of failure to respond, but also difficulties in getting accurate information when funders did respond. We therefore decided to rely only on the policies published on the funders' websites. We also simplified funder eligibility, including funders with at least one RCT funded in the years 2016 to 2018 instead of one RCT funded per year over these 3 years, in order to cover a larger number of funders. Lastly, we added the list of the top 100 pharmaceutical companies in terms of drug sales in 2016 to complete the list of commercial funders. The detailed history of these changes is available on OSF (https://osf.io/ujbf2/).

## Results

### Survey of funders' data-sharing policies

**Funder selection and data extraction.**    Searches and extraction of eligible funders started on 15 February 2019 and ended with a consensus on 10 September 2019. One hundred and

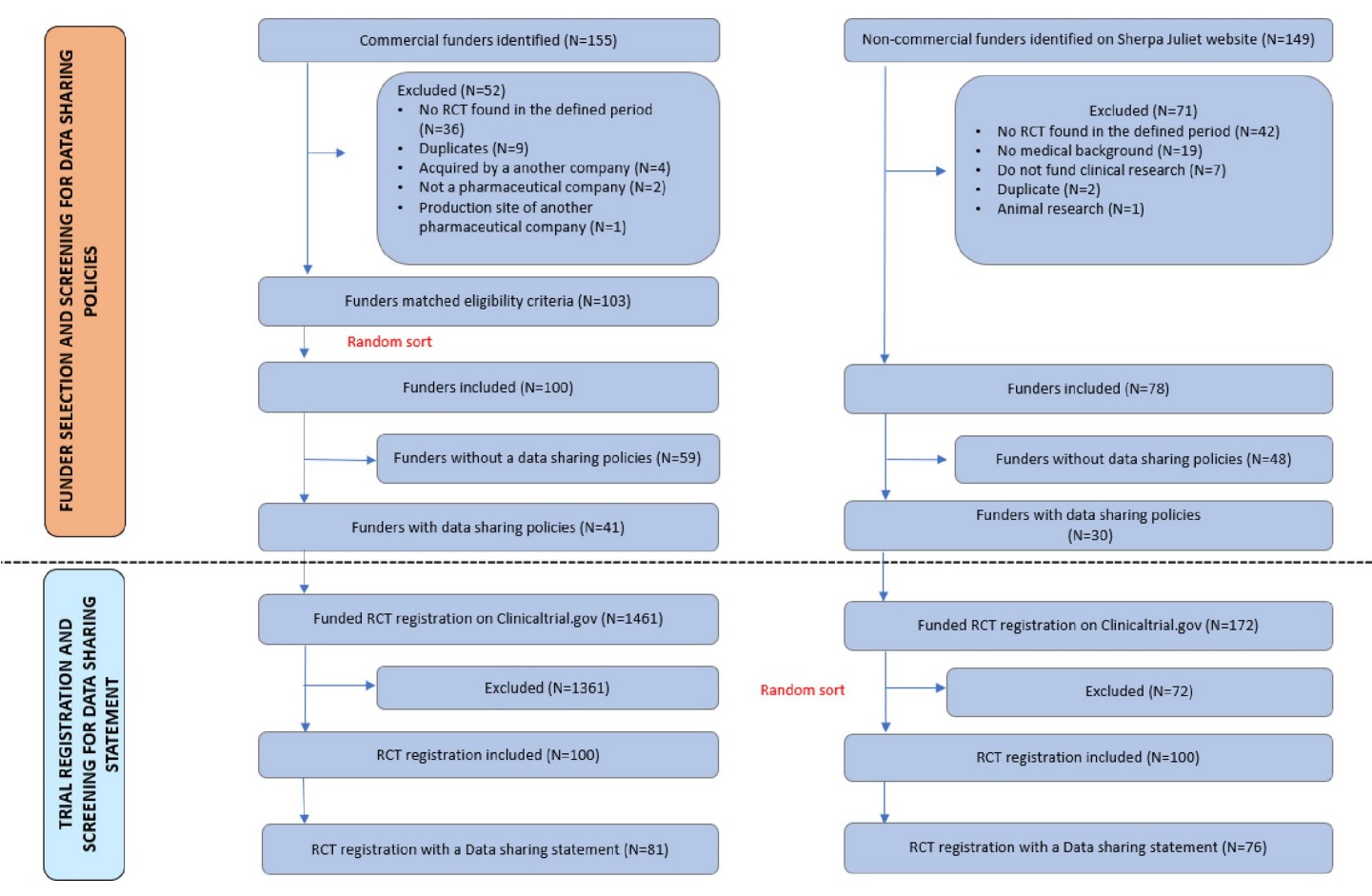

**Fig 1.**

forty-nine non-commercial funders were identified from the Sherpa/Juliet website. Only 78 remained after applying our eligibility criteria and were included. Thirty-five commercial funders were identified from EFPIA, 27 from PhRMA, 37 from IFPMA, 67 from LEEM, along with the top 100 pharmaceutical companies in terms of drug sales, yielding 155 funders without duplicates. 103 of these met our inclusion criteria and 100 were randomly selected for inclusion. Fig 1 details the selection process.

Non-commercial funders. Of the seventy-eight non-commercial funders included, seventy-four (95%) were from high-income countries, two (2.5%) from upper middle-income countries and two (2.5%) were world organizations. Fifty-three funders (68%) were from Europe and central Asia, eighteen (23%) from North America, five (6%) were from east Asia and Pacific countries. The most widely represented countries were the UK (31 funders), the USA (9 funders) and Canada (9 funders). Twenty non-commercial funders (26%) were DORA signatories.

Thirty (out of 78; 38%) non-commercial funders had a data-sharing policy. Table 1 details the characteristics and policies of these funders. Eleven (37%) of the non-commercial funders with a data-sharing policy provided a starting date for their policies. Sixteen (53%) funders asked grant recipients to share data through data-sharing platforms or repositories, one specified the name of the platform (Clinical study data request) and another funder suggested several repositories like Dryad, Dataverse, Figshare and Zenedoo. In terms of sanctions, fifteen

**Table 1. Data-sharing features for funders with a data-sharing policy.**

| | Non-commercial funders (N = 30) | Commercial funders (N = 41) |
|---|---|---|
| **Mention of data sharing policy starting date** | 11 (37%) | 17 (41%) |
| **Data request methods** | | |
| Through data-sharing platform/data enclave/dedicated portal/repositories | 16 (53%) | 30 (73%) |
| By contacting trial investigators | 0 (0%) | 3(7%) |
| **Mention of data-sharing platforms:** | | |
| VIVLI | 0 (0%) | 6(15%) |
| YODA | 0 (0%) | 1(2%) |
| CSDR | 1(3%) | 10(24%) |
| SOAR | 0 (0%) | 1(2%) |
| **Type of shared data or documents mentioned?** | | |
| IPD | 2(7%) | 33(80%) |
| Code | 0 (0%) | 1(2%) |
| Other documents | 1 (3%) | 31 (76%) |
| **Data request review panel** | | |
| Independent | 0 (0%) | 15(37%) |
| Internal review panel | 1 (3%) | 5 (12%) |
| Both | 0 (0%) | 6 (15%) |
| Specialist committee | 0 (0%) | 2 (5%) |
| **Restriction of data availability time?** | 1 (3%) | 2 (5%) |
| **Specify time frames for data-sharing** | 9 (30%) | 0(0%) |
| **Existence of sanctions for non-compliance with policies?** | 15 (50%) | 0(0%) |
| **Reward for sharing data?** | 0 (0%) | 0(0%) |
| **Specific funding for data sharing?** | 13 (43%) | 0 (0%) |
| **DORA Signatory?** | 12 (40%) | 0 (0%) |

Data are presented as numbers (percentages).

(50%) non-commercial funders mentioned that the review of the data-sharing plan was part of the funding decision and that non-compliance can lead to a suspension of the grant or refusal of a future grant application. Thirteen funders (43%) mentioned that they could provide funding to cover data-sharing costs. None of the funders mentioned incentives or rewards for sharing data in their policies.

Eighteen funders (60%) made data-sharing policies mandatory and twelve (40%) funders encouraged data-sharing. Box 1 shows some examples of the policies and Fig 2A provides an

---

## Box 1: Example of supportive policy and mandatory policy

1.Supportive policies

*"**NIH believes that data sharing is essential** for expedited translation of research results into knowledge, products, and procedures to improve human health. NIH endorses the sharing of final research data to serve these and other important scientific goals and expects and **supports** the timely release and sharing of final research data from NIH-supported studies for use by other researchers."* -National Institutes of Health *(NIH)*

> *"**We believe it is important to share clinical trial** data with the public and the scientific community. Sharing improves Research, Knowledge & Patient Care"*- Servier
>
> *"The MRC **expects** valuable data arising from MRC-funded research **to be made available** to the scientific community with as few restrictions as possible so as to maximize the value of the data for research and for eventual patient and public benefit."*—Medical Research Council
>
> 2- Mandatory policies
>
> *"All applicants seeking funding from Parkinson's UK will be **required** to submit a data sharing plan as part of their research grant application. If data sharing is not appropriate, applicants must include a clear explanation why."*–Parkinson's UK
>
> *"It is essential that institutions and PIs share renewable reagents and data developed using Simons Foundation funds with other qualified investigators. PIs will be **required** to have a renewable reagents and data-sharing plan in place prior to receiving a grant"*- Simons Foundation
>
> *"[. . .] All AHRQ-funded researchers **will be required** to include a data management plan for sharing final research data in digital format, or state why data sharing is not possible"*- Agency for Healthcare Research and Quality

overview of the words most frequently used in all the policies. Public funders' policies most often referred to the importance of the data management plan.

Extraction files used for this study and the relevant parts of policies, summarizing funders' positions on data-sharing are available on OSF (https://osf.io/ujbf2/).

**Commercial funders.**   Seventeen (out of 100, 17%) commercial funders included were generic pharmaceutical companies (generic pharmaceutical companies included were those that met our eligibility criteria, and therefore funded at least one clinical trial in the years 2016 to 2018). Ninety were from high-income countries, seven were from lower-middle income countries and three were from upper-middle income countries. The most widely represented countries were the USA (25 funders), Japan (16 funders) and France (13 funders). Forty-four were from Europe and central Asia, twenty-seven were from North America, twenty-two were from east Asia and Pacific countries and seven were from south Asia. None of the commercial funders were DORA signatories.

Forty-one (out of 100, 41%) commercial funders had a data-sharing policy that mentioned their commitment to make clinical trial data available on request (none of them was a generic pharmaceutical company). Thirty-one (out of 41, 76%) of funders with data-sharing policies are members of an organization with established data-sharing guidelines (PhRMA or EFPIA) and one (of 41, 2%) declares that it follows the principles without being a member.

Table 1 details the characteristics and policies of these 41 funders. Seventeen (41%) of them mentioned the starting date of their policies. Thirty (73%) mentioned that they shared their data through a data-sharing platform or a dedicated portal, and three (7%) after a direct contact from the requestor. The remaining funders did not provide details on requests. Clinical Study Data Request was the most widely recommended data-sharing platform (24%). Thirty-three (80%) of the funders mentioned that they made IPD available on request. Thirty-one

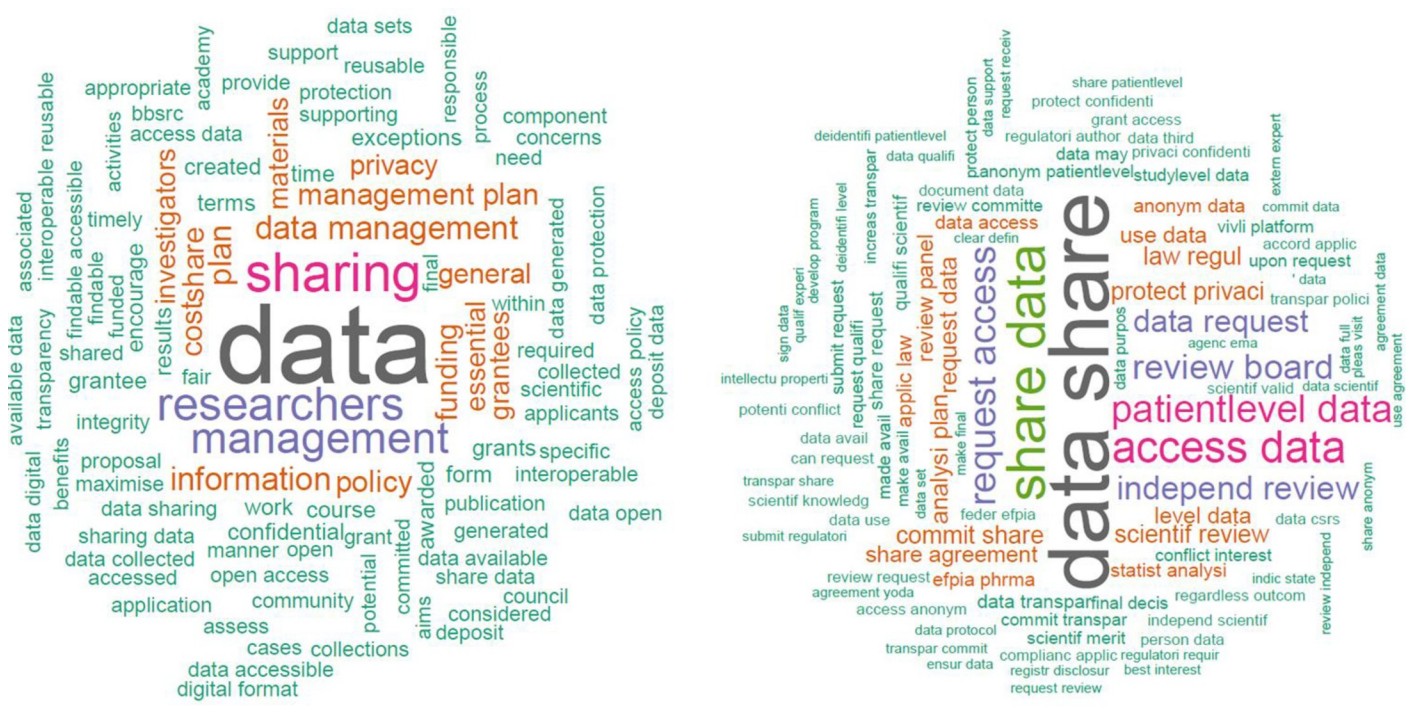

*a: Non-commercials funders*　　　*b: Commercials funders*

**Fig 2. Word cloud illustrating frequent words in funders' policies.**

(76%) specified making other documents besides IPD available (e.g. clinical study reports, study level data, protocols). Concerning the examination of data requests, fifteen (37%) funders mentioned that data requests were evaluated by an independent review panel, five (12%) by an internal review panel, six (15%) mentioned both an internal and an independent panel and two (5%) mentioned a "specialist committee" without further information. Concerning the availability time for the data shared, only two of the funders specified a restriction on the duration of availability (data available for 24 months and data available for 12 months with a possibility for extension). None of the funders mentioned incentives or rewards for sharing-data, sanctions for non-compliance with the policy or funding for data-sharing procedures in their policies.

All commercial funder policies found supported data-sharing. Qualitatively, the distinction between "mandatory" and "encouraging" policies was not applicable because these policies did not apply to an external sponsor but to the commercial funder directly. The policies tended to contain statements supporting trial data-sharing allowing external researchers to request trial data. Fig 2B and Box 1 present a word cloud and some example of these types of policies.

The full data extracted for all funders is available on OSF (https://osf.io/ujbf2/).

## Survey of RCTs funded by funders with a data-sharing policy

**RCT selection and data extraction.** Searches and extraction of eligible RCTs started on 27th September 2019 and ended with a consensus on 8th November 2019. One hundred and seventy-one study registrations on Clinicaltrial.gov were found for seventeen different non-commercial funders with data-sharing policies and six hundred and fifty-seven study

registrations for thirty-six different commercial funders with data-sharing policies. Fig 1 reports the selection process.

Non-commercial funders. The hundred trials randomly selected were funded by fifteen non-commercial funders. The most widely represented funders were NIH (61 trials) and Wellcome Trust (8 trials). A data-sharing statement was present for seventy-seven (77%, 95% IC [67–84%]) registered RCTs funded by non-commercial funders. Among the hundred registrations, 12% [7%-20%] had a "Yes" statement to IPD sharing and 12% [7%-20%] an "Undecided" statement. 12% [7% - 20%] mentioned information about the availability of supporting material such as protocols (11% [6%– 19%]), statistical analysis plans (9% [4%– 17%]) and clinical study reports (6% [2%– 13%]). The time period for data availability was specified in 11% [6%– 19%] of the registrations. Six registrations [2%– 13%] specified that data would be freely accessible and 6% [2%– 13%] specified the methods to have access to data (email or website). Table 2 details these data-sharing statements.

Commercial funders. The hundred trials randomly selected were funded by twenty-seven different funders. The most widely represented funders were Novartis (14 trials), Merck (10 trials) and GSK (10 trials). A data-sharing statement was present for eighty-one (81% [72% - 88%]) registered RCTs funded by commercial funders. Among the hundred registrations, 59% [49%– 69%] has a "Yes" statement to IPD sharing, 9% [4%– 17%] an "Undecided" statement and 16% [10%-25%] a "No" statement (with 2 of them justifying that the reason for not sharing were respectively "the trial meets one or more of the exceptions described" and "individual participants could be re-identified"). 37% [28%– 47%] of RCT registrations mentioned information about the availability of supporting material. 12% [7%– 20%] mentioned that data access would be limited to twelve months and 6% [2%– 13%] that data would be made accessible for "viable scientific projects". Data requests were to be reviewed by an independent panel for 18 [11%– 27%] funders or by a mixed (internal and independent) panel for 2 [0.3%– 8%] funders. Table 2 details these data-sharing statements.

**Table 2. Data-sharing statement details on trials registrations.**

|  | Non-commercial funded RCTs N = 100 | Non-commercial funded RCTs N = 100 |
|---|---|---|
| **Data sharing plan in the registration?** | 77 [67% - 85%] | 81 [72% - 88%] |
| **Intention to share IPD** |  |  |
| Yes | 12 [7% - 20%] | 59 [49%– 69%] |
| Undecided | 12 [7%– 20%] | 9 [4%– 17%] |
| No | 54 [44%– 64%] | 16 [10%– 25%] |
| Nothing specified | 22 [15%– 32%] | 16 [10%– 25%] |
| **Information about supporting material availability?** | 12 [7%– 20%] | 37 [28%– 47%] |
| **Information about protocol availability?** | 11 [6%– 19%] | 33 [25%– 44%] |
| **Information about SAP availability?** | 9 [4%– 17%] | 32 [23%– 42%] |
| **Information about CSR availability?** | 6 [2%– 13%] | 33 [24%– 43%] |
| **Mention of timeframe of data availability?** | 11 [6%– 19%] | 19 [12%– 28%] |
| **Data will be freely accessible?** | 6 [2%– 13%] | 26 [18%– 36%] |
| **Mention of restriction of data access?** | 1 [0.5%– 6%] | 12 [7%– 20%] |
| **Mention of a specific aim for data reuse?** | 3 [0.7%– 9%] | 6 [2%– 13%] |
| **Mention of data access methods?** | 6 [2%– 13%] | 47 [37%– 57%] |

Data are presented as percentages and their corresponding 95% confidence intervals.

## Discussion

We found that 38% of non-commercial funders and 41% of commercial funders had a data-sharing policy in place, as mentioned on their websites. Most of the commercial funders are part of larger organizations (e.g. PhRMA, EFPIA) that have guidelines in place to implement data-sharing, so that commercial funders have more homogeneous attitudes toward data-sharing. In contrast, public funders showed broader heterogeneity in their recommendations. For non-commercial funders with a data-sharing policy, 60% made data-sharing mandatory and 40% encouraged data-sharing. The terms of the policies differ from one funder to another (non-commercial or commercial). Non-commercial funders' data-sharing policies contain recommendations for grant recipients to provide a data management plan and /or follow the FAIR principles, in most cases, as part of recommendations for a funding request. Commercial funder policies are more focused on request and means of access to individual patient data with supporting material (more often for a study in progress or completed) than on planning data-sharing upstream. Often policies lacked certain crucial information, as noted in previous audits [8–10]. For instance, there was a lack of information on the existence of incentives and/ or the type of data request review panel and/or on recommendation of specific platforms for non-commercial funders. Commercial funder policies often lacked information on sanctions and time frames for sharing data.

While we did not directly compare commercial and non-commercial funder enforcement of their policies, it seems that the data-sharing policies were more effectively implemented in data-sharing statements of trials funded by commercial funders: among RCTs registered on Clinicaltrial.gov, 77% and 81% respectively for non-commercial and commercial funders detailed a data-sharing plan, but 12% and 59% respectively expressed an explicit intention to share data. This result is in line with another important aspect of transparency, which is the observation that, despite being far from optimal, commercial funders perform better than non-commercial funders in ensuring availability of individual study results on registers such as clinicaltrials.gov [19, 20]. In addition, the low percentages observed suggest difficulties in implementation of funders' data-sharing policies. These difficulties could result from a lack of understanding the policies [21] or from reluctance in the part of investigators [22]. Planning upstream data-sharing and implementing it after a trial can be challenging.

As our survey points out, funders do not provide for incentives for data-sharing, and funding specially dedicated to data-sharing in not put in place by all of them. This lack of incentives and funding could hinder the implementation of the policies put in place. It is also possible that trialists registering their trial on clinicaltrials.gov do not attach importance to data-sharing plans at the time of registration. Importantly, the registration of a sharing plan (even if the plan was not to share data) was not 100% despite the fact that it has been made mandatory by the ICMJE for publishing an article in its member and affiliated journals. In addition, data-sharing plans were often unclear and some information was contradictory: in some registrations, it was indicated that there was "no plan to share IPD" while details were given about the procedure to access IPD. And indeed, a previous study [21] shared the same concerns and already noted that "several descriptions of IPD sharing plans reflected confusion or uncertainty about the term *IPD* and the meaning of the term *sharing*".

### Comparison with other studies

Estimates in our survey were different from the proportions of funders with data-sharing policies found in previous studies for commercial funders, but in the same range for non-commercial funders. While the methods and exhaustiveness of the previous surveys differed, making any direct comparison difficult, our results suggest that the proportion of funders with a data-

sharing policy has not dramatically improved across the years. For commercial funders, the 2016 estimation [8] of 96% was derived from a sample comprising the 25 biggest companies, and a 2018 survey [23] found that a data-sharing policy was available for 52% of a sample of 61 trials, funded by commercial funders (35 funders). For non-commercial funders, a 2017 survey [24] found 56% of non-commercial funders with a data-sharing policy in a sample of 18 funders.

Bergeris and al [21] examined responses on IPD sharing-related fields on Clinicaltrial.gov and found that 72% of the 35 621 trial records analyzed on August 31, 2017 had responded to the IPD sharing plan. Unlike our study, this study was carried out before the ICMJE requirement [25] and did not explore whether the registered studies included were indeed funded by funders with data-sharing policies. However, it was found that only 36.2% of the studies indicated an intention to share IPD or were undecided whether to share or not.

## Strengths and limitations of the study

We tried to limit the selection bias by exploring a large, diversified, number of funders without focusing on only certain specific funders such as the top pharmaceutical industries. We relied on well-known lists of funders. However, to our knowledge, there is no existing exhaustive list of all possible funders worldwide and therefore a selection bias could persist. For instance, funders on the Sherpa list are mostly from the UK. We performed a similar survey on French funders [18] and found 9/31 (29%) funders with a data-sharing policy, corresponding to 19% (850.032.000 €) of the financial volume of the French funders surveyed. Only 2 of these French funders were also listed in the Sherpa list. Overall, these results suggest that our estimations are still subject to selection bias and could result in a possible overestimation of the number of funders with data-sharing policies. Furthermore, when we assessed registered trials, some funders were overly represented (such as the NIH among the public funders) reflecting the large number of trials they have funded in comparison with other funders.

We tried to limit the information bias by performing an independent extraction by two authors. However, some missing information (e.g. starting date of policy implementation, penalties for non-compliance. . .) was still likely, as the information about data-sharing policies was very poorly structured and heterogeneous across the different websites. A survey contacting the funders directly could have retrieved different information, with however the risk of non-response. For instance, it is possible that data-sharing policies are implemented but not mentioned or only partly described on the funders' websites. And this bias is perhaps less marked for commercial funders like the EFPIA, and PhRMA joint "Principles for Responsible Clinical Trial Data Sharing" [26] stipulates that funders must have information pages dedicated to their data-sharing commitment. Lastly, funders' policies can change and it is likely that some funders that had no explicit policy when we performed our searches have now implemented one.

## Perspectives

The suboptimal performances of funders in setting up and implementing data-sharing policies that we have highlighted in this study call for collective action. Misunderstanding [21], non-adherence [27], or lax application of recommendations are obstacles to consider.

Providing transparent information that reflects funders' commitments and positions toward data-sharing is one of the first important actions that funders can undertake in this direction. As a first step, the creation of an exhaustive list of funders and their policies would enable a continuous and systematic audit of their policies and research outputs.

However, existing policies are heterogenous, especially among non-commercial funders. As with commercial funders, groups of non-commercial funders could together define best practices with an agenda for implementation. It should also be noted that the recommendations for sharing data can be standardized to be applicable to clinical trials funded by commercial and non-commercial bodies [28]. Involving researchers as well as trial participants in the design of best practices of this type is an initiative to be considered by funders, as it would make it possible to identify and address the most important leverages and concerns to consider when implementing effective data-sharing policies.

Moreover, providing evidence of the value of data-sharing will encourage the implementation of more effective policies. Any new policy should have an evaluation component, and we suggest that funders invest in studies on the global impact of their policies on the generation of new knowledge. In addition, any evidence of clinical data-sharing benefits will probably convince the community to adopt data-sharing policies. For instance, interventional trials comparing the impact of various data-sharing policies could explore outcomes such as the production of new knowledge, the enhancement of research reproducibility, and all the different promises of data-sharing. Again, convincing evidence that data-sharing produces the intended results could address the concerns expressed by some trialists [27, 29].

## Conclusion

Funders have a key role to play in making data-sharing a standard in clinical research. Our survey shows that there is room for improvement with regard to their data-sharing policies. We call for a standardization of policies, with a strong evaluation component, to make sure that, when in place, these policies are effective.

## Supporting information

**S1 File.**
(TXT)

## Author Contributions

**Conceptualization:** Jeanne Fabiola Gaba, Florian Naudet.

**Data curation:** Jeanne Fabiola Gaba, Maximilian Siebert.

**Formal analysis:** Jeanne Fabiola Gaba.

**Funding acquisition:** Florian Naudet.

**Investigation:** Jeanne Fabiola Gaba.

**Methodology:** Jeanne Fabiola Gaba, Maximilian Siebert, Alain Dupuy, David Moher, Florian Naudet.

**Project administration:** Florian Naudet.

**Supervision:** Florian Naudet.

**Validation:** Jeanne Fabiola Gaba, Maximilian Siebert, Alain Dupuy, David Moher, Florian Naudet.

**Writing – original draft:** Jeanne Fabiola Gaba.

**Writing – review & editing:** Jeanne Fabiola Gaba, Alain Dupuy, David Moher, Florian Naudet.

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
