## [Decision Letter · Decision Letter 0]

13 Jul 2020

PONE-D-20-10962

Funders’ data-sharing policies in therapeutic research: an audit of commercial and non-commercial funders

PLOS ONE

Dear Dr. GABA,

Thank you for submitting your manuscript to PLOS ONE. After careful consideration, we feel that it has merit but does not fully meet PLOS ONE’s publication criteria as it currently stands. Therefore, we invite you to submit a revised version of the manuscript that addresses the points raised during the review process.

We look forward to receiving your revised manuscript.

Kind regards,

Despina Koletsi, Dipl.D.S, MSc, Dr. med. dent, MSc, DLSHTM, PGCHE

Academic Editor

PLOS ONE

Journal Requirements:

Thank you for stating the following in the Competing Interests section:

'I have read the journal's policy and the authors of this manuscript have the following competing interests:

FN received funding from the French National Research Agency for this project.'

a. Please confirm that this does not alter your adherence to all PLOS ONE policies on sharing data and materials, by including the following statement: "This does not alter our adherence to  PLOS ONE policies on sharing data and materials.” (as detailed online in our guide for authors http://journals.plos.org/plosone/s/competing-interests). If there are restrictions on sharing of data and/or materials, please state these.

Please note that we cannot proceed with consideration of your article until this information has been declared.

b. Please respond by return email with your amended Competing Interests Statement and we will change the online submission form on your behalf.

Additional Editor Comments (if provided):

Reviewers' comments:

Reviewer's Responses to Questions

**Comments to the Author**

1. Is the manuscript technically sound, and do the data support the conclusions?

Reviewer #1: Yes

2. Has the statistical analysis been performed appropriately and rigorously? 

Reviewer #1: Yes

3. Have the authors made all data underlying the findings in their manuscript fully available?

Reviewer #1: Yes

4. Is the manuscript presented in an intelligible fashion and written in standard English?

Reviewer #1: Yes

5. Review Comments to the Author

Reviewer #1: This research aimed to describe commercial and non-commercial funders' data-sharing policies and to assess the

compliance of funded RCTs with the existing data-sharing policies. I feel this article satisfies the PLOS ONE criteria (see below)

1. The study presents the results of original research. Yes

2. Results reported have not been published elsewhere. Yes

3. Experiments, statistics, and other analyses are performed to a high technical standard and are described in sufficient detail. Yes

4. Conclusions are presented in an appropriate fashion and are supported by the data. Yes

5. The article is presented in an intelligible fashion and is written in standard English. Yes

6. The research meets all applicable standards for the ethics of experimentation and research integrity. Yes

7. The article adheres to appropriate reporting guidelines and community standards for data availability. N/A

Overall, the methodology of this study is robust and appears to be consistent with a previous study published by this research team (Data sharing and reanalysis of randomized controlled trials in leading biomedical journals with a full data sharing policy: survey of studies published in The BMJ and PLOS Medicine). The article is well-written and concise. My only suggestion would be the authors consider amending the title of this study from audit to survey. This article is not really an audit as the results are not compared against an evidence based standard.

6. PLOS authors have the option to publish the peer review history of their article (what does this mean?). If published, this will include your full peer review and any attached files.

Reviewer #1: No

---

## [Author Response · Author response to Decision Letter 0]

22 Jul 2020

Dear Reviewer,

We are very grateful for your encouraging comments and valuable feedback on our work. We are delighted to hear that you find our work well-written and concise. 

Thank you for this suggestion to amending the title of the article. We agree with you. The word “audit” has been replaced by the word “survey” in the title and in the main text.

"Funders' data-sharing policies in therapeutic research: a survey of commercial and non-commercial funders"

“As our survey points out, funders do not provide for incentives for data-sharing, and funding specially dedicated to data-sharing in not put in place by all of them. This lack of incentives and funding could hinder the implementation of the policies put in place. It is also possible that trialists registering their trial on clinicaltrials.gov do not attach importance to data-sharing plans at the time of registration.”

---

## [Editor Report · Decision Letter 1]

28 Jul 2020

Funders’ data-sharing policies in therapeutic research: a survey of commercial and non-commercial funders

PONE-D-20-10962R1

Dear Dr. GABA,

We’re pleased to inform you that your manuscript has been judged scientifically suitable for publication and will be formally accepted for publication once it meets all outstanding technical requirements.

Kind regards,

Despina Koletsi, Dipl.D.S, MSc, Dr. med. dent, MSc, DLSHTM, PGCHE

Academic Editor

PLOS ONE

Additional Editor Comments (optional):

All issues raised have been adressed. The manuscript may proceed to publication